# A Three-Phase Model Characterizing the Low-Velocity Impact Response of SMA-Reinforced Composites under a Vibrating Boundary Condition

**DOI:** 10.3390/ma12010007

**Published:** 2018-12-20

**Authors:** Mengzhou Chang, Fangyun Kong, Min Sun, Jian He

**Affiliations:** College of Aerospace and Civil Engineering, Harbin Engineering University, Harbin 150001, China; changmengzhou@hrbeu.edu.cn (M.C.); kongfangyun@hrbeu.edu.cn (F.K.); sunmin@hrbeu.edu.cn (M.S.)

**Keywords:** SMA-reinforced composite, low-velocity impact, vibrating boundary, numerical analysis

## Abstract

Structural vibration induced by dynamic load or natural vibration is a non-negligible factor in failure analysis. Based on a vibrating boundary condition, the impact resistance of shape memory alloy (SMA)-reinforced composites was investigated. In this investigation, a modified Hashin’s failure criterion, Brinson’s model, and a visco-hyperelastic model were implemented into a numerical model to characterize the mechanical behavior of glass fiber/epoxy resin laminates, SMAs, and interphase, respectively. First, a fixed boundary condition was maintained in the simulation to verify the accuracy of the material parameters and procedures by a comparison with experimental data. Then, a series of vibrating boundaries with different frequencies and amplitudes was applied during the simulation process to reveal the effect on impact resistances. The results indicate that the impact resistance of the composite under a higher frequency or a larger amplitude is lower than that under a lower frequency or a smaller amplitude.

## 1. Introduction

Fiber-reinforced composite materials have been widely used and investigated in recent years due to their unique properties, such as high stiffness, high strength, and low density [1]. However, the composites’ applications are limited by their weak impact resistance, especially in unidirectional cross-ply fiber/matrix laminates. Delamination between adjacent layers and debonding between the fiber and the matrix have contributed to the evolution of damage in composites, and this can be explained by the weak interfacial properties [2].

This disadvantage can be overcome by changing the structure of the material; e.g., using a short fiber instead of a long fiber. The mechanical properties of short-fiber-reinforced composites are strongly influenced by manufacturing process factors, such as the injection position, the sample’s geometry, and the pressure and temperature during the molding process [3,4]. Other factors that affect the mechanical properties, such as the fiber location, length, diameter, and orientation, have been studied by Thomason [5,6,7]. The effect of the mould flow direction (MFD) on the interfacial shear strength and the tensile strength of composites has been investigated [8,9]. The results indicate that the tensile strength (and elastic modulus) of samples machined perpendicular to the MFD are nearly 40% lower than that of samples machined parallel to the MFD.

Functional materials and new structures have been used to improve the mechanical properties of composites. Shape memory alloys (SMAs) have been embedded between adjacent layers for a reinforcement purpose considering their shape memory effect; i.e., changing shape (elastic modulus) in accordance with temperature and stress [10,11]. In our previous research, the effect of SMA positions on the damage behavior and impact response (including peak force, displacement, and energy) of laminates subjected to a low-velocity impact have been investigated [12]. In Khalili’s research, the effect of SMA type (wires, plates, strips, tubes, or layers) on dissipation of the impact energies has been studied [13,14]. Shariyat et al. [15,16] have developed a higher-order global–local hyperbolic plate theory aimed at studying the asymmetric displacement fields. Their calculations indicate that SMAs have the ability to change shape, to repair damage, and to improve the impact resistance property of composites [15,16]. Metal layers have been applied in conventional fiber-reinforced polymers to improve the impact resistance [17]. Other structures, such as particle-reinforced [18,19], sandwich plate [20,21], and three-dimensional (3D) fabric [22,23] structures, have been developed in recent years.

We note that a fixed boundary is widely used in experimental analyses. Except for low- and high-velocity impacts, the mechanical properties of a composite under an eccentric impact [24], multiple impacts [25], and compression-after-impact [26] have received much attention due to their role in engineering practices. However, a vibrating boundary condition is unavoidable when membrane structures are subjected to an impact [27] or wind flow [28]. The effect of vibration on the impact resistance of materials has rarely been discussed due to the difficulties associated with real experimental conditions. Only the vibration response during and after an impact can be observed to evaluate the impact resistance and damage state [29,30]. In Pérez’s work, damage to a carbon fiber reinforced polymer (CFRP) induced by a low-velocity impact, and its effect on the vibration response, was investigated by a micro-mechanical approach [31].

Interfacial debonding between the SMA and the matrix, which is one common failure model, is still a key problem in composites [32]. In our previous work, a three-phase model (matrix, reinforcer, and interphase) was introduced to evaluate the mechanical behavior of a fiber-reinforced composite [33,34]. In this paper, this model is further developed to match the SMA-reinforced composites. Based on this model, the effect of vibration on the impact resistance of SMA-reinforced composites is investigated through a series of frequencies and amplitudes.

## 2. The Three-Phase Model

In this model, SMA-reinforced glass fiber/epoxy composites are regarded as having three phases: a glass fiber/epoxy composite laminate phase, an SMA phase, and an interphase between the SMA and the laminate.

### 2.1. Material Property of the Glass Fiber/Epoxy Composite Laminate

Composite laminates containing glass fiber and an epoxy matrix are regarded as anisotropic materials at the macroscale. The constitutive model can be expressed as:
(1)$$\left\{\begin{array}{l} \sigma_{11}\\ \sigma_{22}\\ \sigma_{33}\\ \sigma_{23}\\ \sigma_{31}\\ \sigma_{12} \end{array}\right\}=\left[\begin{array}{cccccc} c_{11} & c_{12} & c_{13} & 0 & 0 & 0\\ c_{21} & c_{22} & c_{23} & 0 & 0 & 0\\ c_{31} & c_{32} & c_{33} & 0 & 0 & 0\\ 0 & 0 & 0 & c_{44} & 0 & 0\\ 0 & 0 & 0 & 0 & c_{55} & 0\\ 0 & 0 & 0 & 0 & 0 & c_{66} \end{array}\right]\left\{\begin{array}{l} \varepsilon_{11}\\ \varepsilon_{22}\\ \varepsilon_{33}\\ \varepsilon_{23}\\ \varepsilon_{31}\\ \varepsilon_{12} \end{array}\right\}$$
where $\varepsilon_{ij}$ and $\sigma_{ij}$ (*i,j* = 1, 2, and 3) are the strain and stress in the *ij*-direction, respectively; $c_{ij}$ are the coefficients of the stiffness matrix, *i,j* = 1, 2,…6. Considering an orthotropic material, the directions are defined as: 1: the fiber’s direction; 2: the in-plane transverse direction of the fiber; and 3: the out-of-plane transverse direction of the fiber.

The stiffness matrix at the linear elastic stage can be expressed as:
(2)$$\begin{array}{c} c_{11}=E_{1}\left(1-\upsilon_{23}\upsilon_{32}\right)\Gamma \\ c_{22}=E_{2}\left(1-\upsilon_{13}\upsilon_{31}\right)\Gamma \\ c_{33}=E_{3}\left(1-\upsilon_{12}\upsilon_{21}\right)\Gamma \\ c_{12}=E_{1}\left(\upsilon_{21}+\upsilon_{31}\upsilon_{23}\right)\Gamma \\ c_{23}=E_{2}\left(\upsilon_{32}+\upsilon_{12}\upsilon_{31}\right)\Gamma \\ c_{13}=E_{1}\left(\upsilon_{31}+\upsilon_{21}\upsilon_{32}\right)\Gamma \\ c_{44}=E_{12}\\ c_{66}=E_{13}\\ \Gamma =1/\left(1-\upsilon_{12}\upsilon_{21}-\upsilon_{23}\upsilon_{32}-\upsilon_{13}\upsilon_{31}-2\upsilon_{21}\upsilon_{32}\upsilon_{13}\right) \end{array}$$
where $E_{i}$(*i* = 1, 2, and 3) is the elastic modulus in the *i*-direction; and $E_{ij}$ and $\upsilon_{ij}$ (*i*, *j* = 1, 2, and 3; i≠j) are the shear modulus and Poisson’s ratio in the *ij*-direction, respectively.

The 3D Hashin’s failure criterion, which accounts for fiber failure and matrix failure, was embedded in ABAQUS using the VUMAT subroutine. In this part, two failure models (tensile failure and compressive failure) were considered for fibers with related damage variables, which can be expressed as [35,36]:

Tensile failure of the fiber:
(3)$$d_{ft}=1:\;\left\{ \begin{array}{l} {\left(\frac{\sigma_{11}}{X_{T}}\right)}^{2}+{\left(\frac{\sigma_{12}}{S_{12}}\right)}^{2}+{\left(\frac{\sigma_{13}}{S_{13}}\right)}^{2}\geq 1\\ \sigma_{11}>0 \end{array}\right.$$

Compressive failure of the fiber:
(4)$$d_{fc}=1:\;\left\{ \begin{array}{l} \left|\frac{\sigma_{11}}{X_{C}}\right|\geq 1\\ \sigma_{11}<0 \end{array}\right.$$
where $X_{T}$ and $X_{C}$ are the tensile strength and compressive strength in the longitudinal direction, respectively; $S_{12}$ and $S_{13}$ are the ultimate shear strength in the 12- and 13-direction, respectively; and $d_{ft}$ and $d_{fc}$ are the damage variables for evaluating the tensile and compressive damage to the fiber, respectively.

Similarly, the tensile failure and compressive failure of the matrix can also be obtained [35,36].

Tensile failure of the matrix:(5)$$d_{mt}=1:\;\left\{ \begin{array}{l} {\left(\frac{\sigma_{11}}{X_{T}}\right)}^{2}+{\left(\frac{\sigma_{12}}{S_{12}}\right)}^{2}+{\left(\frac{\sigma_{22}}{Y_{T}Y_{C}}\right)}^{2}+\frac{\sigma_{22}}{Y_{T}}+\frac{\sigma_{22}}{Y_{C}}\geq 1\\ \sigma_{22}+\sigma_{33}>0 \end{array}\right.$$

Compressive failure of the matrix:
(6)$$d_{mc}=1:\;\left\{ \begin{array}{l} {\left(\frac{\sigma_{11}}{X_{T}}\right)}^{2}+{\left(\frac{\sigma_{12}}{S_{12}}\right)}^{2}+{\left(\frac{\sigma_{22}}{Y_{T}Y_{C}}\right)}^{2}+\frac{\sigma_{22}}{Y_{T}}+\frac{\sigma_{22}}{Y_{C}}\geq 1\\ \sigma_{22}+\sigma_{33}<0 \end{array}\right.$$
where $Y_{T}$ and $Y_{C}$ are the tensile strength and compressive strength in the transverse direction, respectively; and $d_{mt}$ and $d_{mc}$ are the damage variables for evaluating the tensile and compressive damage to the matrix, respectively.

Using the damage variables and the related parameters, the stress is decreased linearly to zero once the failure criterion is reached. The stiffness coefficients obtained from Equation (2) should be recalculated during the damage process, as:
(7a)$$c_{11}=E_{1}\left(1-\upsilon_{23}\upsilon_{32}\right)\Gamma \left(1-d_{f}\right)$$
(7b)$$c_{22}=E_{2}\left(1-\upsilon_{13}\upsilon_{31}\right)\Gamma \left(1-d_{f}\right)\left(1-d_{m}\right)$$
(7c)$$c_{33}=E_{3}\left(1-\upsilon_{12}\upsilon_{21}\right)\Gamma \left(1-d_{f}\right)\left(1-d_{m}\right)$$
(7d)$$c_{12}=E_{1}\left(\upsilon_{21}+\upsilon_{31}\upsilon_{23}\right)\Gamma \left(1-d_{f}\right)\left(1-d_{m}\right)$$
(7e)$$c_{23}=E_{2}\left(\upsilon_{32}+\upsilon_{12}\upsilon_{31}\right)\Gamma \left(1-d_{f}\right)\left(1-d_{m}\right)$$
(7f)$$c_{13}=E_{1}\left(\upsilon_{31}+\upsilon_{21}\upsilon_{32}\right)\Gamma \left(1-d_{f}\right)\left(1-d_{m}\right)$$
(7g)$$c_{44}=E_{12}\left(1-d_{f}\right)\left(1-s_{mt}d_{mt}\right)\left(1-s_{mc}d_{mc}\right)$$
(7h)$$c_{55}=E_{23}\left(1-d_{f}\right)\left(1-s_{mt}d_{mt}\right)\left(1-s_{mc}d_{mc}\right)$$
(7i)$$c_{66}=E_{13}\left(1-d_{f}\right)\left(1-s_{mt}d_{mt}\right)\left(1-s_{mc}d_{mc}\right)$$
where $s_{mt}$ and $s_{mc}$ are the factors that control the reduction in shear stiffness according to the tensile and compressive failure, respectively. The parameters $d_{f}=1-\left(1-d_{ft}\right)\left(1-d_{fc}\right)$ and $d_{m}=1-\left(1-d_{mt}\right)$
$\left(1-d_{mc}\right)$ are the global damage variables characterizing the fiber and the matrix, respectively.

### 2.2. Material Property of the SMA

A similar stress-strain relationship to Equation (1) can be observed if the reinforced material is isotropic elastic, i.e., glass fiber or carbon fiber. However, the constitutive model of SMA is sensitive to temperature and stress. Among all of the models, Brinson’s model [37,38] has been used widely due to its high accuracy. Here, the constitutive law of SMA based on energy balance equations is derived considering the effect of the temperature *T*, and the phase conversion between martensite and austensite can be expressed as:
(8)$$d\sigma_{s}=E_{s}\left(\varepsilon_{s},\xi ,T\right)d\varepsilon_{s}+\Omega \left(\varepsilon_{s},\xi ,T\right)d\xi +\Theta \left(\varepsilon_{s},\xi ,T\right)dT$$
where $\sigma_{s}$ and $\varepsilon_{s}$ are the Piola–Kirchhoff stress and the Green strain of SMA; ξ is the martensite fraction characterizing the phase conversion; and $E_{s}$, Ω, and Θ are the elastic modulus, the transformation coefficient, and the thermal coefficient, respectively.

Some assumptions are made to simplify the application, as follows: the elastic modulus $E_{s}$ is a function of martensite fraction ξ; and the transformation coefficient is also connected with the elastic modulus:
(9a)$$E_{s}\left(\varepsilon ,\xi ,T\right)=E_{s}\left(\xi \right)=E_{a}+\xi \left(E_{m}-E_{a}\right)$$
(9b)$$\Omega \left(\xi \right)=-\varepsilon_{L}E\left(\xi \right)$$
where $E_{m}$ and $E_{a}$ are the elastic moduli for martensite and austenite; and $\varepsilon_{L}$ is the maximum residual strain. The equations of the phase conversion between martensite and austenite can be expressed as (determined by temperature and stress):

(a) Phase conversion to martensite:
(10a)$$\begin{array}{c} \mathrm{If}\;T>M_{s}\;\mathrm{and}\;\sigma_{s}^{cr}+C_{m}\left(T-M_{s}\right)<\sigma <\sigma_{f}^{cr}+C_{m}\left(T-M_{s}\right)\\ \xi_{S}=\frac{1-\xi_{S_{0}}}{2}\cos\left\{\frac{\pi }{\sigma_{s}^{cr}-\sigma_{f}^{cr}}\left(\sigma -\sigma_{f}^{cr}-C_{m}\left(T-M_{s}\right)\right)\right\}+\frac{1+\xi_{S_{0}}}{2} \end{array}$$
(10b)$$\xi_{T}=\xi_{T_{0}}-\frac{\xi_{T_{0}}}{1-\xi_{S_{0}}}\left(\xi_{S}-\xi_{S_{0}}\right)$$
(10c)$$\begin{array}{c} \mathrm{If}\;T<M_{s}\;\mathrm{and}\;\sigma_{s}^{cr}<\sigma <\sigma_{f}^{cr}\\ \xi_{S}=\frac{1-\xi_{S_{0}}}{2}\cos\left\{\frac{\pi }{\sigma_{s}^{cr}-\sigma_{f}^{cr}}\left(\sigma -\sigma_{f}^{cr}\right)\right\}+\frac{1+\xi_{S_{0}}}{2} \end{array}$$
(10d)$$\xi_{T}=\xi_{T_{0}}-\frac{\xi_{T_{0}}}{1-\xi_{S_{0}}}\left(\xi_{S}-\xi_{S_{0}}\right)+\Delta_{T\xi }$$
(10e)$$\Delta_{T\xi }=\left\{ \begin{array}{l} \frac{1-\xi_{T_{0}}}{2}\left[\cos\left(a_{m}\left(T-M_{f}\right)\right)+1\right]:M_{f}<T<M_{s}\;\mathrm{and}\;T<T_{0}\\ 0:\mathrm{else} \end{array}\right.$$

(b) Phase conversion to austenite:
(11a)$$\begin{array}{c} \mathrm{If}\;T>A_{s}\;\mathrm{and}\;C_{a}\left(T-A_{f}\right)<\sigma <C_{a}\left(T-A_{s}\right)\\ \xi =\frac{\xi_{0}}{2}\left\{\cos\left[a_{a}\left(T-A_{s}-\frac{\sigma }{C_{a}}\right)\right]+1\right\} \end{array}$$
(11b)$$\xi_{S}=\xi_{S_{0}}-\frac{\xi_{S_{0}}}{\xi_{0}}\left(\xi_{0}-\xi \right)$$
(11c)$$\xi_{T}=\xi_{T_{0}}-\frac{\xi_{T_{0}}}{\xi_{0}}\left(\xi_{0}-\xi \right)$$
where $M_{s}$ and $M_{f}$ denote the start and finish temperature of the martensite phase, respectively; $A_{s}$ and $A_{f}$ denote the start and finish temperature of the austenite phase, respectively; $C_{m}$ and $C_{a}$ are the material properties describing the relationship between stress and temperature of the martensite phase and the austenite phase, respectively; and $\sigma_{s}^{cr}$ and $\sigma_{f}^{cr}$ represent the critical transformation stress at the start and the finish of the transformation, respectively. In Equations (10) and (11), parameters $a_{m}=\pi /\left(M_{s}-M_{f}\right)$ and $a_{a}=\pi /\left(A_{f}-A_{s}\right)$. The stress-strain relationship at an arbitrary temperature *T* can be obtained according to Equations (8)–(11).

### 2.3. Material Property of the Interphase

In our previous research, the time (or strain rate)-dependent stress-strain relationship of a glass-fiber-reinforced polymer composite has been investigated [33,34]. In this model, the stress $\sigma_{in,ij}\left(t\right)$ of the interphase is a function of strain history, and can be expressed as:
(12)$$\sigma_{in}\left(t\right)=\int_{0}^{t}g\left(t-t^{\prime }\right){\dot{\varepsilon }}_{in}\left(t^{\prime }\right)dt^{\prime }$$
where *g* is the relaxation modulus, $t^{\prime }$ is the new time variable, ${\dot{\varepsilon }}_{in}\left(t^{\prime }\right)=\partial \varepsilon_{in}\left(t^{\prime }\right)/\partial t^{\prime }$ is the strain rate. The strain history $\varepsilon_{in}\left(t\right)$ can be obtained using Boltzmann’s superposition principle. Also, the relaxation modulus g can be expressed using the discrete relaxation spectrum, as follows:
(13)$$g\left(t\right)=g_{0}+\sum_{i=1}^{n}g_{i}e^{\frac{-t}{t_{i}}}$$
where $g_{0}$, $g_{i}$, and $t_{i}$ are the related parameters that can be obtained from relaxation tests. The parameters in the interphase part can be obtained by fitting tensile and pull–put tests with different loading speeds [33,34].

### 2.4. Boundary Condition

For a sample with dimensions $L_{x}\times L_{y}\times L_{z}$ = 100 mm × 100 mm × (n × 0.2 mm), *n* = 16 for our experiment, a fixed boundary is employed to investigate the accuracy of the numerical model. The frequency and the amplitude of the four sides x=0, $x=L_{x}$, y=0, and $y=L_{x}$ are zero, and are shown in Figure 1a.

As for the vibrating boundary condition (Figure 1b), the movement of the boundaries can be expressed as:
(14)z=Asin(2πft)
where *A* and f are the amplitude and the frequency of the vibration, respectively.

## 3. The Effect of the Fixed Boundary Condition on Impact Resistance

The fixed boundary condition is kept in the experimental process.

### 3.1. Composite Laminates

Simulation results of the model under a fixed boundary were investigated and compared with the experimental results. The stacking sequence of the laminate is [0°,90°]_8_; 0° and 90° are the glass fiber’s layer angles in the X-direction. The sample ($L_{x}\times L_{y}\times L_{z}$ = 100 mm × 100 mm × 3.2 mm) was subjected to an impact from a rigid half-ball cylinder at the center of the top surface, as shown in Figure 1. The half ball’s diameter is 14 mm, and its mass is 8 kg (Steel). Two impact energies were considered in the tests (32 J and 64 J), and the corresponding impact velocities are 2.83 m/s and 4 m/s, respectively.

The material parameters were obtained by fitting with the experiments [12,34], as shown in Table 1.

SMA wires (diameter, 0.2 mm) were embedded in the middle layer laminate (between layer 8 and layer 9) at a distance of 5 mm (with a total of 21 wires for a model). In summary, four types of experiments have been conducted, as shown in Table 2.

### 3.2. Simulation Result: Damage State During the Impact Process

The model has been built in ABAQUS according to the details mentioned above, and the meshed result is shown in Figure 2. In ABAQUS, the ‘Rigid body’ element type of impactor was chosen; and the ‘C3D8R’ element type of laminate and interphase was chosen. The total number of elements in this model was 102,996. In the center region, the mesh net was optimized to ensure the accuracy of the model. The ‘Explicit’ step (*t* = 0.01 s), integrated into VUMAT (the constitutive relations of laminate, SMA, and interphase), was chosen to simulate the impact process.

**Group A1**: The simulation results on the composite laminate under an impact at different times (0.0015 s, 0.0030 s, 0.0045 s, 0.0060 s, 0.0075 s, and 0.0090 s) are shown in Figure 3 and Figure 4. Under the lower impact energy (32 J), a generally elastic behavior of the composite laminate can be expressed as: the deformation is increased with time *t* from the initial state (*t* = 0) to the maximum deformation (*t* ≈ 0.0045 s), then the deformation is decreased.

As shown in Figure 3, it is clear that the impactor has been bounced back by the composite laminate. Three layers—layer 16, layer 8, and layer 1—have been chosen to investigate the fracture morphology during the impact, as shown in Figure 4. A hole-shaped damage region can be found on several layers, especially layer 16; however, the damage region on layer 8 and layer 1 is smaller, as shown in Figure 4.

**Group A2**: Different from group A1, the composite is destroyed completely under the higher impact energy (64 J), as shown in Figure 5. During the simulation process, the impactor was found to move along the top layer to the bottom layer of the composite without being bounced back (*t* > 0.0030 s). This can also be confirmed by the experimental result, as shown in Figure 5g. The velocity of the impactor was reduced in the breakdown process and then remained as a constant.

From Figure 6c, a larger damage region can be found at the final state. This is different from group A1, and can be explained by delamination due to the friction between the impactor and layer 1 during penetration.

**Groups A3 and A4**: Embedding SMA alloys is an effective way to improve the impact resistance of composite laminates. As shown in Figure 7a, the SMA was stretched to a larger strain in the case of 32 J. In Figure 7b, a broken or invalid state of the SMA is obtained due to the larger strain, which is beyond the critical value. More specifically, five SMAs in the center region were chosen to demonstrate the working mechanism, as shown in Figure 8c,d.

Beyond that, the defect is obvious: the damage region of layer 8 (contact with the SMA) is larger than that of group A1 and A2, as shown in Figure 8a,b.

### 3.3. Simulation Result: Absorbed Energy and Contact Force

Two important items—absorbed energy and contact force—were obtained from ABAQUS and compared with the experimental data, as shown in Figure 9. The relative errors of energy comparison are 5.7%, 6.3%, 13.8%, and 7.3% for groups A1–A4, respectively. As for the force-time curves, more inflection points can be found on the simulation curves due to the breaking and removal of the element. Even so, the tendency of the simulation curves agrees well with the experimental curves. The relative errors of the force comparison are 7.3%, 8.1%, 8.9%, and 9.2% for groups A1–A4, which can also be accepted.

From the comparison in this section, the accuracy of the material parameters and the accuracy of the model used in ABAQUS can be tested. Furthermore, the constitutive relationship of SMA and the processing of the interface (between the SMA and the laminate) are also regarded as appropriate. This is the foundation of the numerical simulation in Section 4.

## 4. The Effect of the Vibrating Boundary Condition on Impact Resistance

The simulation of the model under the vibrating boundary condition under impact is investigated in this section. The same model ($L_{x}\times L_{y}\times L_{z}$ = 100 mm × 100 mm × 3.2 mm, 16 layers) as shown Section 3.1 was subjected to an impact by a rigid half-ball cylinder with a fixed energy of 32 J at the center of the top layer. Two types of composites have been investigated: without and with SMA.

### 4.1. The Effect of Amplitude

In order to fully understand the influence of amplitude on the impact resistance, a low frequency *f* = 1000 cycles/s was maintained (10 cycles during the simulation process, *t*_tot_ = 0.01 s). Several amplitudes *A* were chosen for the study, as shown in Table 3.

The positive value in Table 3 means that the movement direction of the boundary is contrary to the impactor’s movement direction (+z direction) at the initial state. The negative value means that the movement directions of the impactor and the boundary are the same at the initial time.

When applying an amplitude *A* to control the movement of the boundary, the morphologies of the composite laminate after impact are shown in Figure 10 and Figure 11. As shown in Figure 10a, the damage region is close to the size of the impactor. From Figure 10a–h, the damage regions are increased as the value of the applied amplitude increases; however, general damage with a larger area can be observed in case of *A* = 0.016 m, as shown in Figure 10g. It is interesting that the damage state depends on the absolute value of the amplitude rather than the value when comparing related groups, e.g., Figure 10c,d. A similar conclusion can be obtained for group C from the simulation results shown in Figure 11.

The details of the impact process for the half model in group B are shown in Figure 12. From Figure 12a, a representative impact process is shown: gradual damage with time *t*, similar to group A1. With increasing amplitude, more elements in the center region have been removed due to the large deformation (or severe vibration), as shown in Figure 12b–d. As for the larger amplitude, 0.016 m, a larger damage region is observed, and even separation from the main model, *t* = 0.001 s, as shown in Figure 12d. More importantly, the time at which the clear damage region can be observed has changed from *t* = 0.005 s to 0.0002 s due to the relative movement. The effect of the value of an amplitude can be further validated by comparing Figure 12b,e.

Three amplitudes are shown to demonstrate the impact process of group C: *A* = 0.0032 m, 0.008 m, and 0.016 m, as shown in Figure 13. From Figure 13a, layer damage along the SMA direction can be found during the impact process. This is due to the weak impact resistance of the laminate after being separated from the SMA, especially for layer 8. With increasing amplitude, a clear delamination can be observed between layer 8 and layer 9, *t* = 0.0005 s, as shown in Figure 13b. As for group C4, clear damage is shown at an early time, *t* = 0.0004 s. The damage state continues to extend even after separation due to the vibrating of the boundary, as shown in Figure 13c.

With different amplitudes, the absorbed energy and contact force are plotted against time, as shown in Figure 14. For group B1 and C1, the maximum value of absorbed energy is 32 J; however, the values at *t* = 0.01 s are 30.04 J and 28.6 J, respectively. As the value of the amplitude increases, the absorbed energy is decreased. For group B3, B5, and B7, the maximum value of absorbed energy is 9.24 J, 4.24 J, and 0.30 J, respectively. The maximum value of absorbed energy in the related opposite-direction groups, i.e., B3–B4, B5–B6, and B7–B8, is maintained at same level and about 20% lower. For the related group C, the absorbed energy is 3.5 times larger. For the high amplitude, the absorbed energy is close to zero, as in group B7 and B8. From *t* = 0 to 0.004 s, the effect of vibrations on the energy-time curve can be observed, and the energy is kept constant, as shown in Figure 14a. In Figure 14b, the force is also affected by the vibrations, and shows more dramatic changes when compared with Section 3.

### 4.2. The Effect of Frequency

In order to fully understand the influence of frequency on the impact resistance, a small amplitude *A* = 0.0032 m was maintained considering the small influence of this level. Several frequencies *f* were chosen for the study, as shown in Table 3.

During the simulation, *f* = 100 cycles/s to 500 cycles/s in groups D1 to D3, the damage states are close to each other at *t* = 0.01 s, and only a small hole can be found according to the fracture morphology shown in Figure 15a and Figure 16b,c. As for the higher frequency, *f* = 2000 cycles/s to 10,000 cycles/s, the damage states show randomicity with a larger area, as shown in Figure 15d,e.

Damage states of SMA-reinforced composite laminates are shown in Figure 15f–j. Overall, the effect of frequency on the damage state is similar to that in group D. It should be noted that the separated region in group E5 retains a more complete shape, as shown in Figure 15j.

Applying different frequencies *f* to the boundary, the simulation process of two groups, D5 and E5, were investigated to demonstrate the details of the damage morphologies of the composite laminate, as shown in Figure 16. For group D4, a hole-shaped damage region gradually appears at time *t* = 0.0025 s, which increases with the impact process, and, at time *t* = 0.01 s, the center region is damaged completely. For group E5, delamination is observed, except for in the hole-shaped damage region. This is mainly due to the SMA’s global enhancement effect.

The relationship between absorbed energy and time or contact force and time can be found in Figure 17. As shown in Figure 17a, the absorbed energy is decreased as the frequency increases. Considering the very low frequency *f* = 100/s, the maximum value of absorbed energy for group D1 and E1 is same: 32 J. Considering the high frequency *f* = 10,000/s, the absorbed energy for group E5 is 2.36 J, and for D5 the absorbed energy is 1.79 J, which means that the composites can barely bear the impact. As shown in Figure 17b, the maximum value of force in the case of *f* < 2000 cycle/s is in the range of 7–7.5 N for group D and 9–12 N for group E. For group D5 and group E5, a saltation is observed when comparing the maximum value of force between adjacent groups, which is mainly due to the transient change in velocity.

More important, the maximum value of absorbed energy and contact force for SMA-reinforced composites is generally 15–30% larger than that of pure glass-fiber-reinforced composites under the same amplitude or frequency.

### 4.3. Statistical Analysis of the Damage State

In Table 4, the maximum energy, *E*_max_, the energy at time 0.01 s, *E*_t = 0.01_, the maximum force, F_max_, and the average force, *F*_avg_, of different groups are shown. An *E*_max_ = 32 J denotes a rebound behavior of the impactor. The average force is defined as:
(15)$$F_{avg}=\frac{\sum_{i=1}F_{i}}{N}$$
where *N* is the total number of output data of force *F*_i_ within a time of 0.01 s. More importantly, the average value is still in accordance with *A* and *f*.

In Figure 18, the ratio between the damaged area and the whole model is plotted against time. Generally, the damaged areas of laminates for *A* < 0.0032 m or *f* < 500 cycles/s are kept at <5%. In Figure 18a, the damaged areas of the top layer (layer 16) for *A* > 0.0032 m or *f* > 500 cycles/s are kept at 30–50%. Differently, the damaged areas of the middle layer (layer 8) and the bottom layer (layer 1) under the same conditions are kept at 30–70% and 30–65%, respectively.

In Figure 19a, the average damage area was investigated by calculating 16 layers, and the results indicate a 4% average damage area for a small amplitude and frequency and nearly a 50% average damage area for a large amplitude and frequency at time *t* = 0.01 s. Figure 19b shows the damage state of the SMA. Damage to the SMA can be observed at an earlier time than damage to the composite laminates, which means that SMA has advantages for absorbing energy.

### 4.4. Mathematical Expression: Effect of Amplitude and Frequency

In this section, the relationship between velocity and amplitude (or frequency) is investigated. The results show a clearly inverse proportion, as follows:
(16a)$$\Delta V=\frac{k_{a}}{\sum_{i=0}m_{i}A^{i}}\left(\mathrm{for}\;\mathrm{given}\;f\right)$$
(16b)$$\Delta V=\frac{k_{f}}{\sum_{i=0}n_{i}f^{i}}\left(\mathrm{for}\;\mathrm{given}\;A\right)$$
where *k*_a_ and *m*_i_ are parameters related to amplitude; and *k*_f_ and *n*_i_ are parameters related to frequency. Inserting Equation (16) into the energy equation, the relationship between absorbed energy and amplitude (or frequency) is obtained as follows:
(17)$$E=\frac{1}{2}mV_{0}^{2}-\frac{1}{2}m{\left(V_{0}-\frac{k}{\sum_{i=0}m_{i}A^{i}\sum_{i=0}n_{i}f^{i}}\right)}^{2}$$
where *E* and *m* are the absorbed energy and the mass of the impactor, respectively. Considering that $F_{avg}=m\times \Delta v/\Delta t$, the average force is also obtained as:(18)$$F_{avg}=\frac{k}{t_{tot}\sum_{i=0}m_{i}A^{i}\sum_{i=0}n_{i}f^{i}}$$

The simulation results using the abovementioned equations are shown in Figure 20 (‘+’ and ‘−’ denote the values of amplitudes; ‘Sim’ denotes the simulations in Section 4; and ‘Eq’ denotes ‘equation’). In this simulation, *i* = 3 was chosen to simplify the expression. The comparison indicates that both the energy and force can be predicted using Equations (17) and (18).

## 5. Conclusions

A three-phase model has been proposed to investigate the effect of a vibrating boundary on the impact resistance of shape memory alloy (SMA)-reinforced composite laminates. Some conclusions have been obtained based on the simulation results under different amplitudes and frequencies.

(1)Under a smaller amplitude (*A* < 0.0032 m) and a lower frequency (*f* < 500 cycles/s), the absorbed energy and contact force of composite laminates are similar to that under a fixed boundary condition. In contrast, both a high frequency and a high amplitude can weaken the impact resistance of composite laminates, where extensive damage can be observed rather than a hole-shaped damage region.(2)The absolute value of amplitude has a greater influence on the impact resistance than the movement direction of the laminates at the initial time. The absorbed energy and contact force in the positive direction are about 20% larger than that in the negative direction.(3)Embedding an SMA can improve the impact resistance of composite laminates due to the superelasticity. In this study, embedding an SMA was found to increase the absorbed energy and contact force by about 15–30%. Also, embedding an SMA can change the damage morphology with respect to shape and proportion.

The proposed three-phase model can provide a tool for efficiently investigating the damage behavior of SMA-reinforced composite laminates.

## Figures and Tables

**Figure 1 materials-12-00007-f001:**
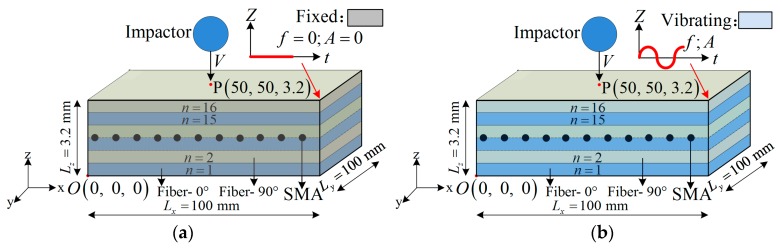
A schematic of the shape memory alloy (SMA)-reinforced composite samples: (**a**) Fixed boundary; (**b**) Vibrating boundary.

**Figure 2 materials-12-00007-f002:**
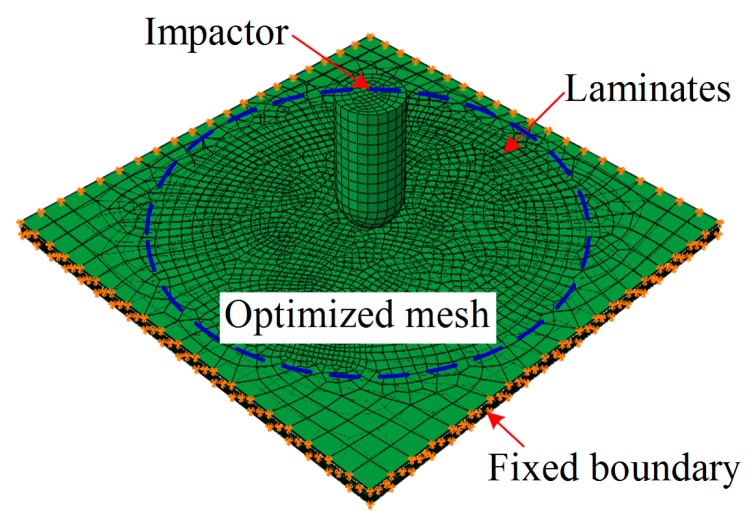
Modeling the impact test of the composite laminate.

**Figure 3 materials-12-00007-f003:**
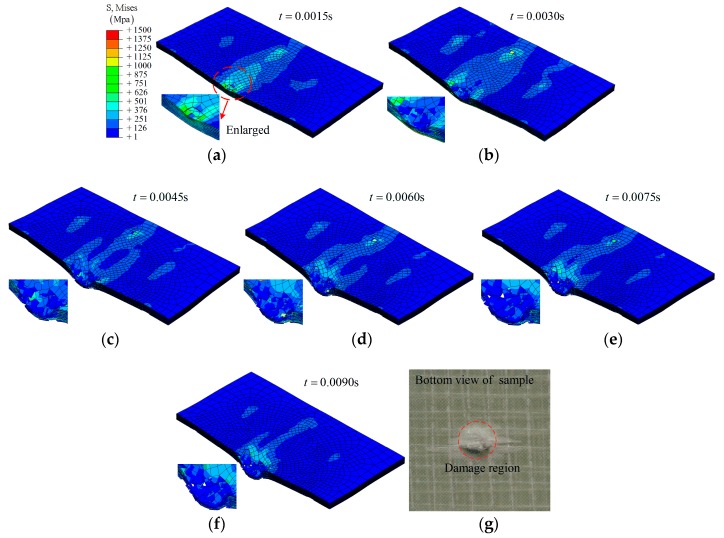
The simulation results on the composite laminates at different times (group A1), and the bottom view of the sample after impact: (**a**) *t* = 0.0015 s; (**b**) *t* = 0.0030 s; (**c**) *t* = 0.0045 s; (**d**) *t* = 0.0060 s; (**e**) *t* = 0.0075 s; (**f**) *t* = 0.0090 s; (**g**) bottom view of sample.

**Figure 4 materials-12-00007-f004:**
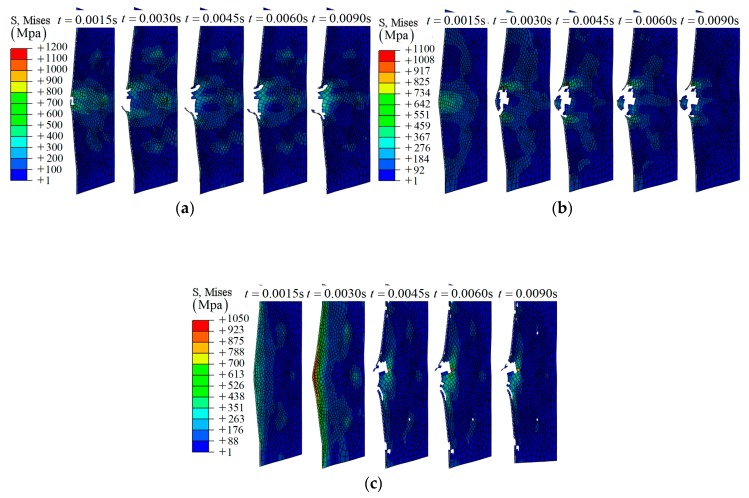
The middle section of different layers during the impact, group A1: (**a**) layer 16; (**b**) layer 8; (**c**) layer 1.

**Figure 5 materials-12-00007-f005:**
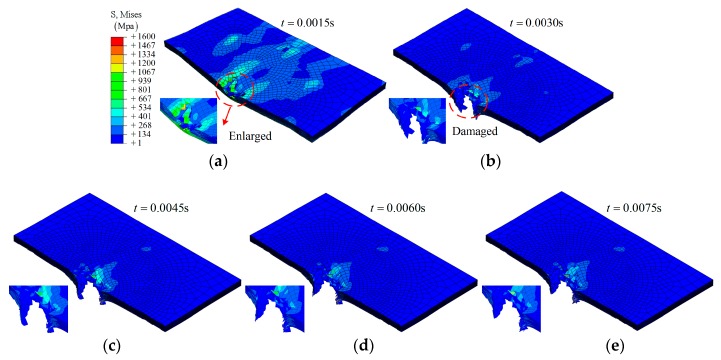
The simulation results on the composite laminates at different times, group A2: (**a**) *t* = 0.0015 s; (**b**) *t* = 0.0030 s; (**c**) *t* = 0.0045 s; (**d**) *t* = 0.0060 s; (**e**) *t* = 0.0075 s; (**f**) *t* = 0.0090 s; (**g**) bottom view of sample.

**Figure 6 materials-12-00007-f006:**
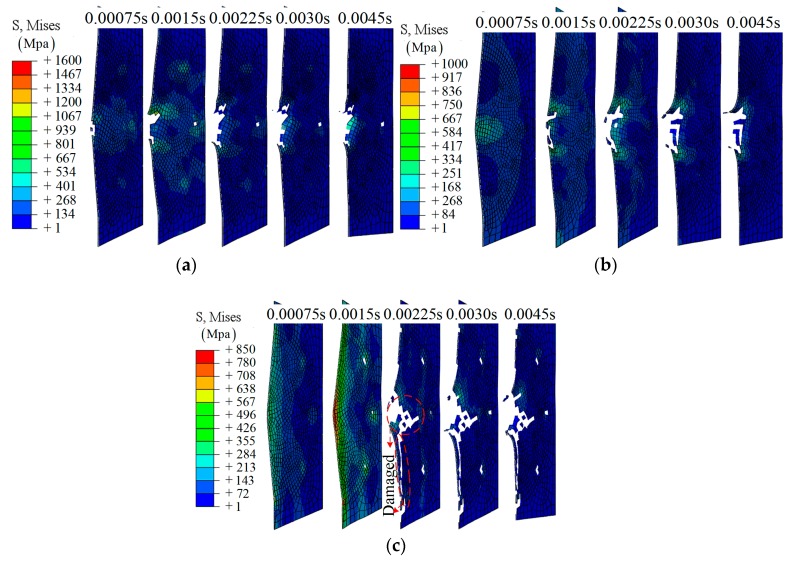
The middle section of different layers during the impact, group A2: (**a**) layer 16; (**b**) layer 8; (**c**) layer 1.

**Figure 7 materials-12-00007-f007:**
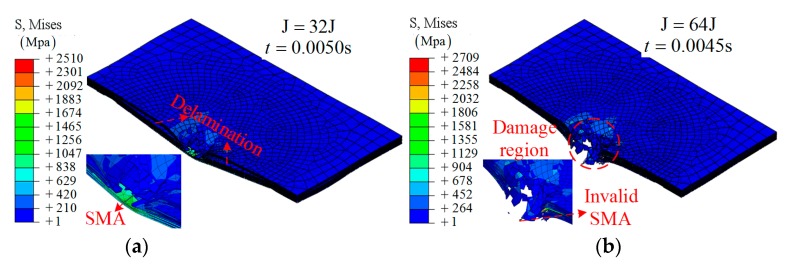
The simulation results on the SMA-reinforced composite laminates at different times: (**a**) group A3, *t* = 0.0050 s; (**b**) group A4, *t* = 0.0045 s.

**Figure 8 materials-12-00007-f008:**
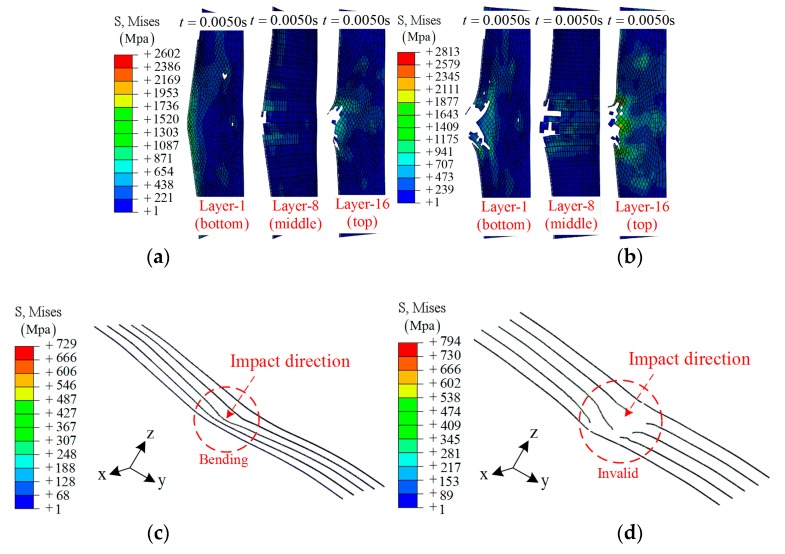
The middle section of different layers during the impact: (**a**) layers in group A3; (**b**) layers in group A4; (**c**) the SMA in group A3; (**d**) the SMA in group A4.

**Figure 9 materials-12-00007-f009:**
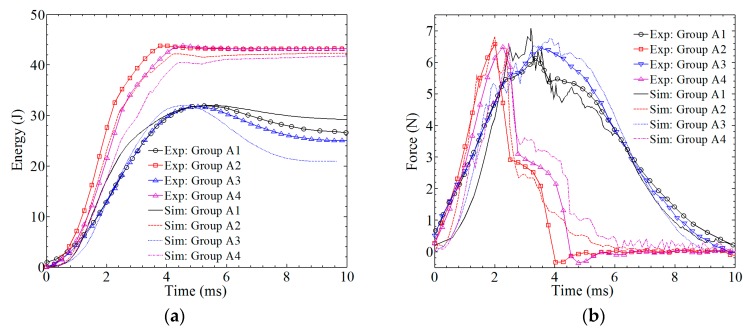
A comparison between the simulation results and the experimental results. (**a**) absorbed energy–time history; (**b**) contact force–time history.

**Figure 10 materials-12-00007-f010:**
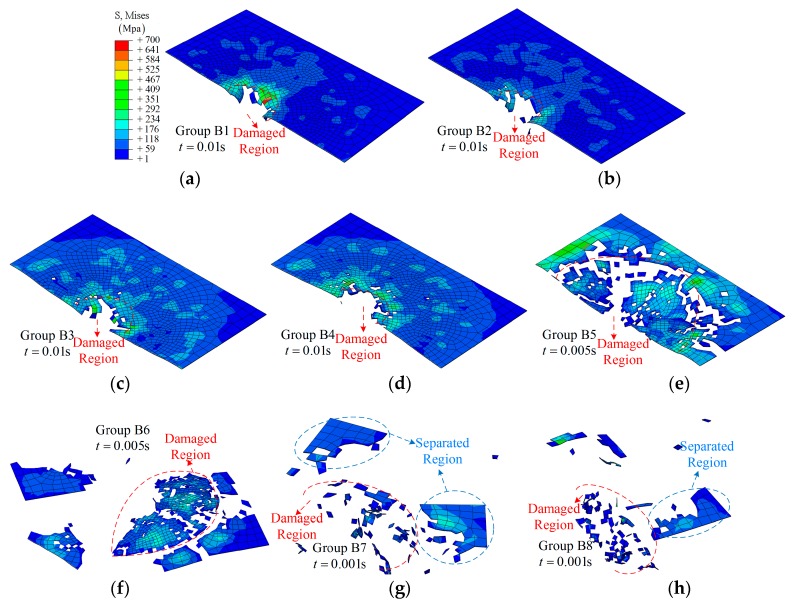
The fracture morphology of the top layer of the composite laminate under different amplitudes: (**a**) Group B1; (**b**) Group B2; (**c**) Group B3; (**d**) Group B4; (**e**) Group B5; (**f**) Group B6; (**g**) Group B7; (**h**) Group B8.

**Figure 11 materials-12-00007-f011:**
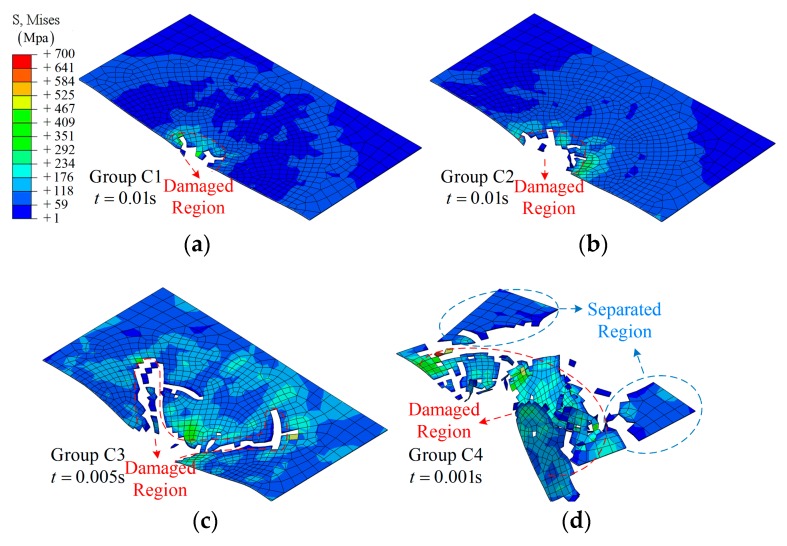
The fracture morphology of the top layer of the SMA-reinforced composite laminate under different amplitudes: (**a**) Group C1; (**b**) Group C2; (**c**) Group C3; (**d**) Group C4.

**Figure 12 materials-12-00007-f012:**
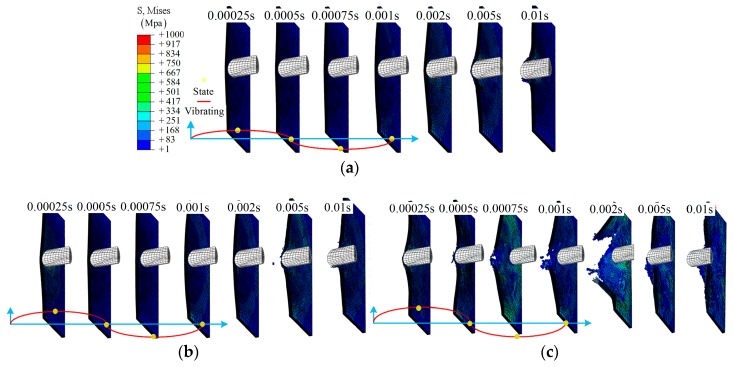
The middle section of the half model during the impact: (**a**) group B1; (**b**) group B3; (**c**) group B5; (**d**) group B7; (**e**) group B4.

**Figure 13 materials-12-00007-f013:**
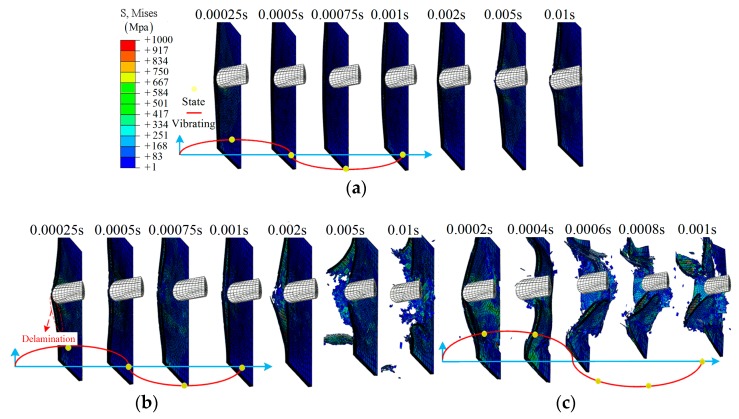
The middle section of the half model during the impact: (**a**) group C2; (**b**) group C3; (**c**) group C4.

**Figure 14 materials-12-00007-f014:**
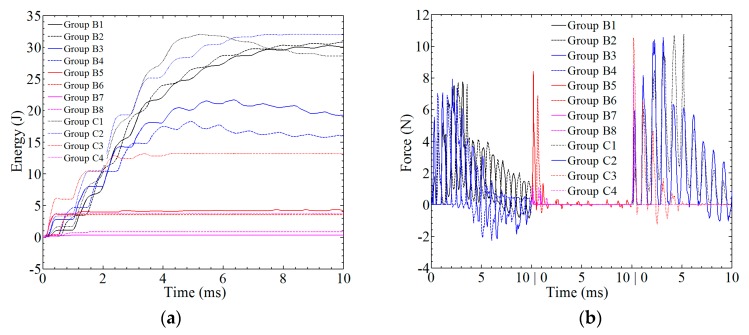
The analysis of the impact resistance of the composite laminate in group B and C: (**a**) absorbed energy; (**b**) contact force.

**Figure 15 materials-12-00007-f015:**
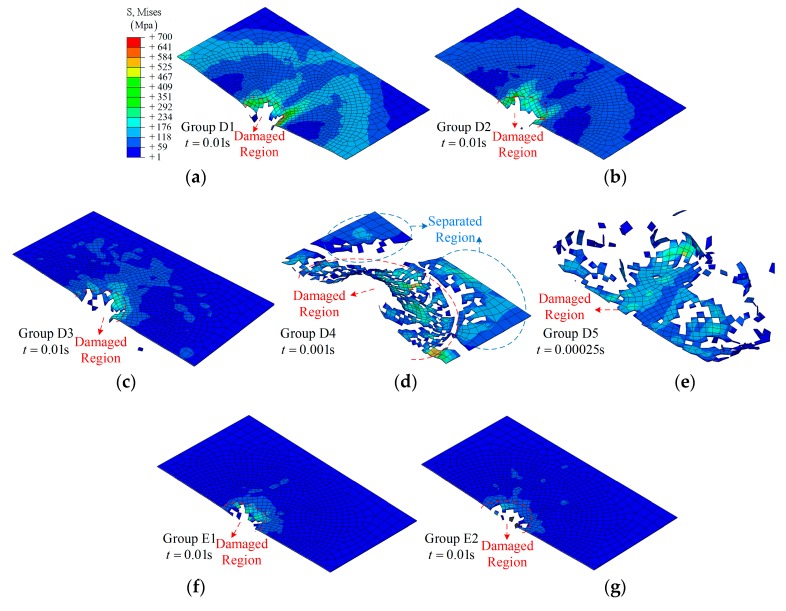
The fracture morphology of the top layer with different frequencies: (**a**) Group D1; (**b**) Group D2; (**c**) Group D3; (**d**) Group D4; (**e**) Group D5; (**f**) Group E1; (**g**) Group E2; (**h**) Group E3; (**i**) Group E4; (**j**) Group E5.

**Figure 16 materials-12-00007-f016:**
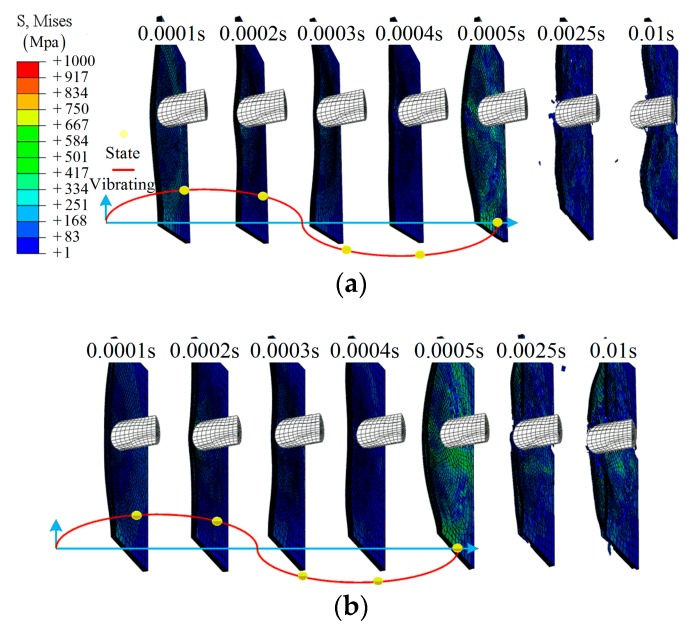
The cross-section of the half model during the impact: (**a**) group D4; (**b**) group E4.

**Figure 17 materials-12-00007-f017:**
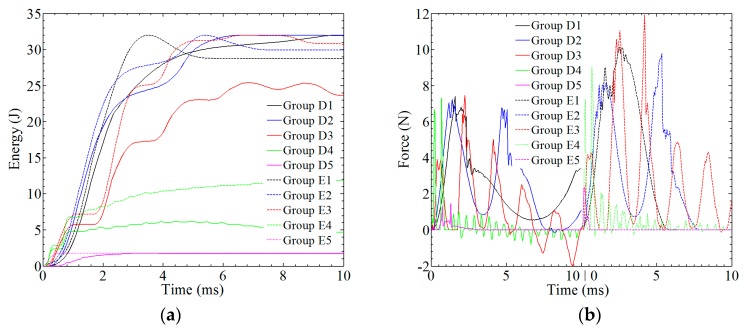
The analysis of the impact resistance of the composite laminate in group D and group E: (**a**) absorbed energy; (**b**) contact force.

**Figure 18 materials-12-00007-f018:**
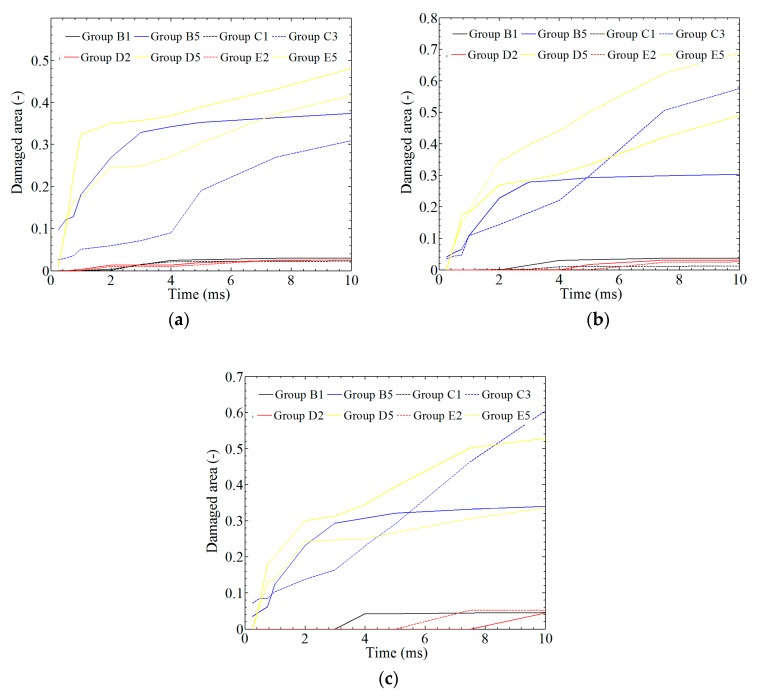
The statistics of the damage area in different groups: (**a**) layer 16; (**b**) layer 8; (**c**) layer 1.

**Figure 19 materials-12-00007-f019:**
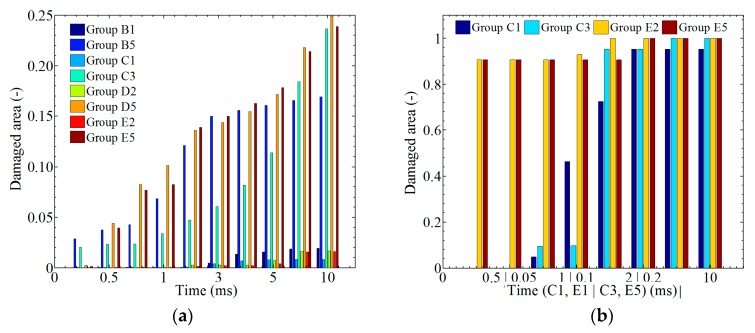
The statistics of the general damage area: (**a**) laminate; (**b**) SMA.

**Figure 20 materials-12-00007-f020:**
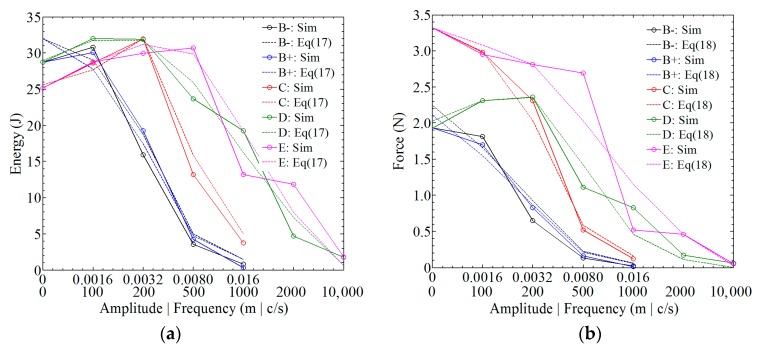
A comparative study of the statistical results and the simulation results: (**a**) *E*_t = 0.01_; (**b**) *F*_avg_.

**Table 1 materials-12-00007-t001:** The material parameters of unidirectional glass fiber/epoxy composite laminates.

Mechanical Constants	Values
Young’s modulus/GPa (*E*_1_, *E*_2_, *E*_3_)	55.2, 18.4, 18.4
Poisson’s ratio ($\upsilon_{12}$, $\upsilon_{13}$, $\upsilon_{23}$)	0.27, 0.27, 0.43
Shear modulus/GPa (*E*_12_, *E*_13_, *E*_23_)	13.8, 13.8, 13.8
Ultimate tensile stress/MPa (*X*_T_, *Y*_T_, *Z*_T_)	1656, 73.8, 73.8
Ultimate compressive stress/MPa (*X*_C_, *Y*_C_, *Z*_C_)	1656, 91.8, 91.8
Ultimate shear stress/MPa (*S*_12_, *S*_13_, *S*_23_)	117.6, 117.6, 117.6

**Table 2 materials-12-00007-t002:** The experimental groups.

	Stacking Sequence	Impact Energy/J
Group A1	[0°,90°]_8_	32
Group A2	[0°,90°]_8_	64
Group A3	[(0°,90°)_4_,SMA, (0°,90°)_4_]	32
Group A4	[(0°,90°)_4_,SMA, (0°,90°)_4_]	64

**Table 3 materials-12-00007-t003:** The amplitudes and frequencies used in the simulation.

	Group B1		Group B2		Group B3		Group B4		Group B5		Group B6		Group B7		Group B8	
*A* (m)	0.0016		−0.0016		0.0032		−0.0032		0.008		−0.008		0.016		−0.016	
*f* (c/s)	1000															
SMA Stacking	NO [0°,90°]_8_															
	**Group C1**				**Group C2**				**Group C3**				**Group C4**			
*A* (m)	0.0016				0.0032				0.008				0.016			
*f* (c/s)	1000															
SMA Stacking	YES [(0°,90°)_4_, SMA, (0°,90°)_4_]															
	**Group D1**			**Group D2**			**Group D3**			**Group D4**				**Group D5**		
*A* (m)	0.0032															
*f* (c/s)	100			200			500			2000				10,000		
SMA Stacking	NO [0°,90°]_8_															
	**Group E1**			**Group E2**			**Group E3**			**Group E4**				**Group E5**		
A (m)	0.0032															
*f* (c/s)	100			200			500			2000				10,000		
SMA Stacking	YES [(0°,90°)_4_, SMA, (0°,90°)_4_]															

**Table 4 materials-12-00007-t004:** The statistics of absorbed energy and contact force.

	A1	B1	B2	B3	B4	B5	B6	A7	B8
*E*_max_ (J)	32	32	32	21.67	18.27	4.30	3.72	0.30	0.80
*E_t_*_= 0.01s_ (J)	28.70	30.04	30.79	9.24	5.90	4.24	3.56	0.30	0.80
*F*_max_ (N)	7.04	7.74	7.69	7.92	7.35	8.39	6.86	0.86	1.13
*F*_avg_ (N)	1.93	1.70	1.81	0.83	0.65	0.16	0.13	0.01	0.02
	**A3**	**C1**		**C2**		**C3**		**C4**	
*E*_max_ (J)	32	32		32		13.20		3.74	
*E_t_*_= 0.01s_ (J)	25.14	28.60		31.97		13.17		3.74	
*F*_max_ (N)	6.89	10.77		10.56		10.52		8.71	
*F*_avg_ (N)	3.32	2.98		2.31		0.52		0.12	
	**D1**	**D2**		**D3**		**D4**		**D5**	
*E*_max_ (J)	32	32		25.38		6.21		1.75	
*E_t_*_= 0.01s_ (J)	31.98	31.92		23.68		4.69		1.74	
*F*_max_ (N)	7.40	7.23		7.45		7.29		1.44	
*F*_avg_ (N)	2.31	2.36		1.11		0.17		0.06	
	**E1**	**E2**		**E3**		**E4**		**E5**	
*E*_max_ (J)	32	32		32		11.86		1.79	
*E_t_*_= 0.01s_ (J)	28.77	29.94		30.70		11.86		1.79	
*F*_max_ (N)	10.15	9.79		11.92		9.02		2.36	
*F*_avg_ (N)	2.95	2.81		2.69		0.46		0.05	

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
