# Peer review of "A Three-Phase Model Characterizing the Low-Velocity Impact Response of SMA-Reinforced Composites under a Vibrating Boundary Condition"

_materials, 2018, doi:10.3390/ma12010007_

Round 1
Reviewer 1 Report
The topic of the article is within the scope of the journal.
Please find attach a pdf with plenty of comments.
English must be improved. I highlighted some things, but a thorough read and review must be done.
Figures are not clear in general. Rainbow colormap is not good for BW publication. In general fontsize is small. Also you vary the limits shown in each figures complicating a direct comparison of the results, make them fix.
The numerical approach is poorly described. Please revise according to the pdf comments. It is not clear to me why you described a homogeneous isotropic elastic material which you seem not to use. Also you present some things within a small strain theory framework and then you seem to change to a finite strain one. You have not talked about mesh sensitivity nor time increment influence in your calculations.
I do not recommend publishing the article in its current form. I do think is interesting but it needs some clarification and improvement.

Author Response
Reviewer-1
The topic of the article is within the scope of the journal.
Please find attach a pdf with plenty of comments.
English must be improved. I highlighted some things, but a thorough read and review must be done.
RE: We have checked the text in paper.
Figures are not clear in general. Rainbow colormap is not good for BW publication. In general fontsize is small. Also you vary the limits shown in each figures complicating a direct comparison of the results, make them fix.
RE: The figures in this manuscript have been modified to make it readable.
The numerical approach is poorly described. Please revise according to the pdf comments. It is not clear to me why you described a homogeneous isotropic elastic material which you seem not to use. Also you present some things within a small strain theory framework and then you seem to change to a finite strain one. You have not talked about mesh sensitivity nor time increment influence in your calculations.
RE: This is a common misconception. In fact, the homogeneous isotropic elastic material can only be used in epoxy matrix. As for the ‘laminate’ part in ABAQUS, an orthotropic model is applied. The stress strain relationships of ‘SMA’ and ‘Interphase’ (Eq.(8)- Eq.(13)) are imbedded into VUMAT.
I do not recommend publishing the article in its current form. I do think is interesting but it needs some clarification and improvement.
RE: Thanks for your comments, it helped me a lot.
Some problems in YOUR PDF:
(1) About references:
RE: We have update some references, please check the highlight part in ‘References’ PART.
(2) The models
RE: We have modified the models and equations, please see ‘2. The three phase model’. We also upload the files RF.33 and RF.34, in which the details of the model and the parameters calculation process can be founded.
In section 2.3, the express of the model is right. In our previous work RF.33, the expression of viscoelastic model is different from Eq.(12) and Eq.(13), however, the essence is same.
(3) ABAQUS model
RE: More details about the numerical model can be founded in LINE 197-202.
(4) Expressive methods of laminates
RE: the expression has modified: subscript ‘8’ denotes the repetition.
(5) ‘More important, the maximum value of absorbed energy and contact force for SMA reinforced composites is generally 15%-30% larger that than of pure glass fiber reinforced composites under same amplitude or frequency.’ in LINE 421-423.
RE: This is based on the statistical results, e.g., group B V.S. group C in TABEL 4.
(6) ‘The absolute value of amplitude has greater influence on the impact resistance rather than the moving direction of laminates at initial time. The absorbed energy and contact force in positive direction is about 20% larger than that in negative direction.’ In LINE 494-496.
RE: This is also based on the statistical results, e.g., group B1 V.S. group B2 in TABEL 4.

Reviewer 2 Report
Please revise your introduction by highlighting fixed and vibrating boundary condition with references and their findings.
Figure 10 and 11: Please make bigger picture to see your sentence.
Author Response
Reviewer-2
Please revise your introduction by highlighting fixed and vibrating boundary condition with references and their findings.
Figure 10 and 11: Please make bigger picture to see your sentence.
RE: We have modified the ‘Introduction’ PART and figures. Please see the related highlight part and figures.
Thanks for your advices.

Round 2
Reviewer 1 Report
The authors have addressed many comments from the first review. Some of them however were not responded.
I attach a pdf with a few comments.
I think a "response to reviewers" letter responding explicitly to every comment is necessary.

Author Response
Reviewer-1
The authors have addressed many comments from the first review. Some of them however were not responded.
I attach a pdf with a few comments.
I think a "response to reviewers" letter responding explicitly to every comment is necessary.
RE: Thanks for your comments. The problems are explained as follows:
Line 75-76: … where epsilon_ij and sigma_ij are the strain and stress tensors, i,j = 1, 2 and 3; c_ij is the stiffness matrix, i,j = 1, 2…6…
RE: The expression of Eq. (1) is appropriate, same as the Eq.(3) in Rf.[17] (upload as 001.jpg).
We have modified the text, as ‘where epsilon_ij and sigma_ij (i,j = 1, 2 and 3) are the strain and stress in ij-direction, respectively; c_ij are the coefficients of stiffness matrix, i,j = 1, 2…6.’
Line 89: Eq.(3)
RE: The explanation of Eq.(3) has been added in Line 92 ‘S_12 and S_13 are the ultimate shear strength in 12- and 13-direction, respectively’.
Line 122-131: Eq.(8)- Eq.(9)
RE: Piola-Kirchhoff stress and Green strain are generally used for metallic materials (elastic-plastic). The temperate related part is not considered in the calculation due to the room temperature kept in the experiments (dT=0).
The stress-strain relationship is considered as a function of martensite fraction- zeta. Eq.(9b) is a commonly used and can be explained by the related part in Rf. [38], as shown in 002.jpg and 003.jpg.
Line 151:…critical transformation stress…
RE: The critical transformation stresses are used to characterize the start and finish of conversion of the martensitic variants, more details can be founded in 004.jpg.
Line 155-165: According to our research…different loading speed [33-34].
RE: We have modified the expression of Line 155-156, as ‘In our research, the time (or strain rate) depended stress-strain relationship of glass fiber reinforced polymer composite has been investigated [33-34].’
In Eq.(12), the time t (upper limit ) is regarded as a constant value; the new time variable t^/ (between 0 and t) is different from t, the explain can be founded in 005-009.jpg. [Shaw M T, MacKnight W J. Introduction to polymer viscoelasticity. John Wiley & Sons, 2018.]
Line 175-178: Eq.(14)
RE: In this paper, the vibration of composite in Z-direction (thickness direction) is discussed. The vibration in x-y plane or fitting external excitation (wind or water flow, etc) is more complex and need further investigated in our future work. Considering that a lot of work has been conducted, the simulations (in Z-direction ) may be suitable for one paper.
Line 180: The fixed boundary condition is kept in the experiment process.
RE: Considering the experimental condition, ONLY fixed boundary can be used. In section 3, the impact resistance of composite under fixed boundary is investigated to lay a foundation of section 4.
LINE 183: The expression of composite laminates:[0^o, 90^o]_8
RE: The expression of laminates is expression as: [0^o, 90^o]_8. That equals to the follows: [0^o, 90^o, 0^o, 90^o, 0^o, 90^o, 0^o, 90^o, 0^o, 90^o, 0^o, 90^o, 0^o, 90^o, 0^o, 90^o] (16 layers).
In the following text, [(0^o, 90^o)_4, SMA, (0^o, 90^o)_4] equals as:
[0^o, 90^o, 0^o, 90^o, 0^o, 90^o, 0^o, 90^o, SMA, 0^o, 90^o, 0^o, 90^o, 0^o, 90^o, 0^o, 90^o] (16 layers and SMA).
This expression is widely used and can be founded in 010 and 011.jpg [Asadi A, Raghavan J. Model for prediction of simultaneous time-dependent damage evolution in multiple plies of multidirectional polymer composite laminates and its influence on creep. Composites Part B: Engineering, 2015, 79: 359-373. Sun M, Wang Z, Yang B, et al. Experimental investigation of GF/epoxy laminates with different SMAs positions subjected to low-velocity impact. Composite Structures, 2017, 171: 170-184.]
Line 208: …at different time (0.0015s<t< 0.009s)…
RE: The step time in ABAQUS is 0.01s. The simulation results at different time (0.0015<t<0.009s) are shown in Fig. (3) (no 0<t<0.01s). We have modified the expression as: The simulation results of the composite laminate under impact at different time (0.0015s,0.0030s, 0.0045s, 0.0060s, 0.0075s and 0.0090s) are shown in Figure 3 and Figure 4.
